# Effect of Different Minerals on Water Stability and Wettability of Soil Silt Aggregates

**DOI:** 10.3390/ma15165569

**Published:** 2022-08-13

**Authors:** Agnieszka Adamczuk, Angelika Gryta, Kamil Skic, Patrycja Boguta, Grzegorz Jozefaciuk

**Affiliations:** Department of Physical Chemistry of Porous Materials, Institute of Agrophysics, Polish Academy of Sciences, Doswiadczalna 4 Str., 20-290 Lublin, Poland

**Keywords:** contact angle, kinetics, aggregate stability, apparent hydrophobicity, soil minerals, wettability, water drop penetration time

## Abstract

Knowledge on the effects of minerals on soil water stability and wettability is mostly gained from experiments on natural soils of different mineral composition. To gain a “clearer” picture, the water stability and wettability of artificial aggregates composed of soil silt and various proportions of pure minerals: kaolinite, montmorillonite, illite, zeolite and goethite, were examined. The wettability was attributed to contact angles measured goniometrically and to the water drop penetration time (WDPT). The water stability was measured by monitoring of air bubbling after aggregate immersion in water and the shrinking sphere model was used to analyse aggregates’ destruction kinetics. The rate of aggregate destruction in water increased with increasing mineral content and it slightly decreased for aggregates composed of all pure minerals except goethite. An apparent hydrophobicity period (a period where the bubbling stopped for some time), resulted most probably from the wavy shape of pores, was observed mainly for aggregates with low mineral proportions. Among all studied minerals, kaolinite increased the water contact angle and water repellency to the greatest extent. With increasing the mineral content in the aggregates up to 8%, contact angles decreased and then increased. Contact angles did not correlate with aggregates’ stability. Aggregates more rapidly penetrated by water (shorter WDPT) were destroyed faster. Water stability of aggregates containing all minerals except illite appeared to be higher for the more mechanically resistant aggregates.

## 1. Introduction

Correct soil structure is a key factor for managing natural resources in sustainable agriculture to maintain high productivity while taking care of the natural environment [1,2]. Soil structure consists of solids of different size and composition, and spaces between them that form clusters of various sizes, i.e., aggregates [3]. The bonding forces inside them are stronger than those between the adjacent aggregates [4]. Soil aggregates affect the content and location of elements, interactions between solid and liquid phases, heat capacity and flows of mass and energy in soil [3,5,6]. Water stability of the aggregates is one of the most important factors determining soil fragments’ release and transportation by runoff and splash, and their re-deposition, surface sealing, pore plugging and the movement (hydraulic conductivity) and storage of soil water. The above processes influence soil health and functionality, biological activity, root penetration, crusting, erodibility, compaction, aeration and carbon sequestration [7,8]. Therefore, modification of soil water stability is used to reactivate the proper functioning of soil [9,10,11].

The decomposition mechanisms of aggregates in water include: slaking, breakdown by differential swelling (montmorillonite swells when water is absorbed, which increases its volume, illite and kaolinite are stronger and nonexpendable), mechanical breakdown by raindrops and physiochemical dispersion, all of which lead to the formation of smaller aggregates or primary particles [12,13,14,15,16]. As far as processes of dispersion of pure minerals into suspensions are well-recognised, the same processes in soils remain not fully understood due to the complex effects of different factors. The ones that increase water stability the most are soil organic matter content (including biological life), amorphous and crystalline aluminium and iron oxides, type and concentration of cations and carbonates, all of which have been most often mentioned in the literature [9,11,15,17,18,19,20,21]. Less attention has been given to parent material, texture and clay mineralogy, which affect, to different extents, soil strength, aggregation and plasticity [22]. The majority of clay minerals exhibit hydrophilic properties [23,24], and therefore clay additions have been used as an effective way of reducing water repellency in sandy soils. Laboratory investigations have shown that the efficiency of clay additions may depend on their mineralogy. During repeated wetting and drying cycles, the clays dominated by kaolinite were much more effective in reducing water repellency than their montmorillonite-rich counterparts [25,26]. However, in sand in which repellency was artificially induced by cetyl alcohol, montmorillonite was more effective than kaolinite [27]. According to Six et al. [28], the presence of kaolinite increased the soil stability. Aminiyan et al. [29] proved that aggregate stability assessed by the mean weight diameter increased with the additions of nanozeolite and zeolite. Zang et al. [30] explored the influence of montmorillonite on improving the water condition in sandy soil. The authors reported that montmorillonite-enriched siltstone increased the available water capacity.

Multiple synergic and/or antagonistic interactions between all the above factors in soils can modify their individual influences to a great extent, and therefore there is a need to study the effects of pure soil components on water stability. Moreover, the structural differences between natural soil aggregates, coming even from very close neighbourhoods, lead to measurements of quite different water stabilities, and studies of the aggregates made from homogenised material are preferred [31]. The impact of pure humic acids, silica, iron oxides and aluminium oxides on the water stability of artificially prepared homogeneous aggregates was studied by Jozefaciuk and Czachor [32]. In this paper, the individual effects of different minerals: kaolinite, illite, montmorillonite, zeolite and goethite, on the water stability of homogeneous soil silt aggregates were studied.

The most common techniques for studying aggregate water stability include the dispersion ratio and the wet-sieving method [33]. These methods consist of three steps: wetting, measurement of the destroyed residue and description of the results in terms of a size distribution or an index of aggregation [34]. These methods fail for rapidly braking aggregates. In the present experiments, a newer method based on monitoring of air bubbling after aggregate immersion [31], which yields precise results regardless of the rate of aggregate destruction, was applied.

## 2. Materials and Methods

### 2.1. Materials

Silt fraction (2–50 µm) composed mainly from feldspars and quartz, extracted from the upper 0–10 cm layer of a Haplic Luvisol, described in detail by Lipiec et al. [35], was mixed with powdered forms of the following minerals:Goethite 71063-100G (Sigma-Aldrich, St Louis, MO, USA),Kaolinite containing <5% illite and ~10% quartz,Illite containing ~10% kaolinite and ~5% quartz,Montmorillonite K10 (Sigma Aldrich Chemie GmbH, Steinheim, Germany),Zeolite, coming from a clinoptilolitic tuff deposit in Sokirnitsa, Ukraine.

Carefully homogenised distilled water-saturated pastes were prepared from the mixtures of the silt and the minerals. The content of particular minerals in the mixtures was 0 (control SILT), 2%, 4%, 8%, 16%, 32%, 64% and 100%. Spherical aggregates of 20 mm in diameter were formed from the pastes using ordinary silicon forms sold in fishing stores to prepare fish-bait. The aggregates were air-dried until constant mass was reached at laboratory conditions (relative humidity around 60% and temperature around 20 °C). A detailed description of the materials, including specific surface area, particle diameter, zeta potential, micropore volume, average pore diameter, solid phase density, surface fractal dimension and volumetric porosity, is presented in [36].

### 2.2. Studies of the Aggregates

The wettability of the aggregates was estimated based on the observation of a four-microliters drop of distilled water settled on a surface of the pressed pellet (4 tons cm^−2^) made from the aggregate material. Specac Atlas Manual Hydraulic Press 15011 (Fisher Scientific, Hampton, NH, USA) was used to prepare the pellets. The video film registering the droplet behaviour was taken in five replicates using a DSA 100 automatic drop shape analyser (KRUSS, Hamburg, Germany). Two parameters were measured: contact angle, α (degs) [37], and water drop penetration time (*WDPT*) (s) [38]. Since contact angles changed in time due to the droplet spreading and soaking, as it is exemplary shown in Figure 1, the initial contact angle measured just after the droplet deposition (t = 0) was assumed to be the most reliable.

From the bulk density (BD) of the aggregates, their composition (%_SILT_ and %_MIN_) and the particle density (PD) of particular aggregate components presented in [36], the total volume of air present in the studied (2 cm diameter) aggregates was calculated:*total air volume* = [1 − BD × (%_SILT_/100/PD_SILT_ + %_MIN_/100/PD_MIN_)] × ^4^/_3_Π (1)

Alternatively, the total volume of air present in the studied aggregates was estimated from their porosities measured by mercury intrusion porosimetry (MIP) using the Autopore IV, 9500 (Micromeritics INC, Norcross, GA, USA) porosimeter.

The water stability of the aggregates was estimated from air bubbling after immersion using a method described by Józefaciuk et al. [31], which is briefly outlined below. The aggregate is thrown into a vessel submerged in water and hanging on a scale pan, and a time-dependent increase in the aggregate weight, Δ*w* (kg), due to the evolution of entrapped air from the interior of the destructed aggregate (decreasing in buoyancy) is registered. The above method was developed for soil aggregates, for which the dependence of Δ*w* on time afforded a characteristic sigmoidal curve, reaching a plateau at the moment when the aggregate is totally destroyed. The above curve fitted very well to a shrinking sphere model:1 − [1 − Δ*w*/(*w_final_* − *w_initial_*)]^1/3^ = k _*_*t*,(2)
where *w_final_* (kg) is the weight of the submerged aggregate after termination of the destruction and *w_initial_* (kg) is its initial weight registered just after immersion, *t* (s) is the time of the destruction process, and k (1/s) is a constant related to the time needed to complete the aggregate destruction, *t*_d_ (s), by the formula:k = 1/*t*_d_. (3)

Defining Δ*w*/(*w_final_* − *w_initial_*) for convenience as an extent of the destruction, α, Equation (2) becomes: 1 − (1 − α)^1/3^ = k _*_*t*.(4)

The data plotted in coordinates of Equation (4) present a straight line, reaching the value of 1 − (1 − α)^1/3^ = 1 at the point where *t* = *t*_d_ (and α = 1).

The simulated curve illustrating the above “pure” destruction process, arbitrarily setting 1/*t*_d_ = 0.01 s^−1^ and Δ*w* = 200 mg, is presented in Figure 1, along with its plot in coordinates of Equation (4).

As this was observed in preliminary experiments for the silt-mineral aggregates studied in this paper (see also Section 3), their destruction (bubbling) curves were much more complex. For some curves, just after the aggregate immersion, a rapid process of initial bubbling was observed that can be attributed to flooding of open conical pores, for which both openings are located on the aggregate surface. The respective curve was simulated as the sum of the “pure destruction” curve from Figure 1 and the initial steep line (fast bubbling) starting at *t* = 0 and terminated after around five seconds (Figure 2a). In the plot of the latter curve in coordinates of Equation (3), one may distinguish a straight line occurring in a zone of the aggregate destruction (Figure 2b). The slope of the aggregate destruction line in Figure 2b differs from that in Figure 1b, which leads to an erroneous destruction time calculation. Fast initial bubbling data should be eliminated to gain the correct value. The amount of air released during fast flooding of funnel pores (*FP*) is:*FP* = *w_final_* [1 − (1 − *INT*)^3^], (5)
where *INT* is the intercept of the straight line in the destruction zone in Figure 2b.

Subtracting *FP* from the aggregate destruction plus the funnel pores’ flooding curve, one obtains a new curve, describing the aggregate destruction process only, as it is depicted in Figure 2c. The slope of the latter curve plotted in coordinates of Equation (4) (Figure 2d) equals the slope in Figure 1b. Now, the correct destruction time is calculated.

The next occurrence observed after fast wetting of the silt-mineral aggregates was a period of no bubbling that was called the apparent hydrophobicity period, *AHP* [s]. It was assumed that this period starts just at the beginning of the aggregate immersion. The simulated curve depicting funnel pores’ flooding and the apparent hydrophobicity period is shown in Figure 3a.

Again, the slope of the aggregate destruction line in Figure 3b differs from the slope in Figure 1b, and the correct destruction time is calculated after subtracting the data of the fast initial bubbling (Figure 3c) from their plot in coordinates of Equation (4) (Figure 3d).

The value of the destruction time depends both on the characteristics of the aggregated material and the size of the aggregate. In the shrinking sphere model, the destruction time is proportional to the initial aggregate surface (*S*_0_). Therefore, the ratio of *t*_d_/*S*_0_ (s m^−2^), which can be read as the time necessary to destroy the unit surface of the aggregate, characterizing the aggregated material regardless of the aggregate size, is used as a water stability parameter.

For measurements of the destruction curves of the studied silt-mineral aggregates, EXPLORER^®^ ANALYTICAL EX324M balance provided by OHAUS (Parsippany, NY, USA) with time resolution adjusted to the given aggregate destruction time was used. The final data presented for aggregate destruction time and the apparent hydrophobicity period are averages from at least six most similar destruction curves selected from ten replicates for each aggregate. Such selection minimises the effects of structural artefacts influencing the destruction. The value of *S_0_* was estimated for each aggregate from its mass divided by bulk density (assumed to be the same for aggregates from each experimental variant).

## 3. Results and Discussion

Exemplary water destruction (bubbling) curves of silt aggregates containing different proportions of minerals are shown in Figure 4.

The bubbling curve of pure silt aggregate seems to be influenced by all three mechanisms occurring during fast wetting: initial fast flooding of funnel pores followed by the period of no bubbling, and finally, the aggregate destruction. Similar shapes of the curves occur for 2–8% concentrations of montmorillonite and zeolite, 2–16% concentrations of kaolinite and goethite and for 2–64% concentrations of illite. Some curves seem to be governed by the pure destruction process only (all 100% mineral aggregates, all 64% mineral aggregates excluding illite and 34% kaolinite and zeolite aggregates). The other curves reflect a coincidence of aggregate destruction and a shorter or longer period of funnel pores’ flooding. Individual approaches to each of the destruction curves, as described in Section 2 (eliminating funnel pores’ flooding, estimation of apparent hydrophobicity period, see Figure 1, Figure 2 and Figure 3), showed that within the detected aggregate destruction zone, the shrinking sphere model for water destruction of the studied aggregates (Equation (4)) was well-applicable. Taking *t*_d_ values calculated from the slopes of the linear fits of the destruction data plotted in coordinates of Equation (4) and dividing them by the initial surface of each aggregate, *S_0_*, the values of the time necessary to destroy the unit surface of the aggregate, *T_d_* = *t*_d_/*S*_0_ (s m^−2^), were calculated. The results are presented in Figure 5.

The water stability of silt-goethite aggregates increases with an increase in the mineral content. For kaolinite and montmorillonite, the water stability increases with the addition of the minerals and slightly decreases for 100% pure minerals. The water stability of zeolite-containing aggregates increases up to 32% of the mineral content and then falls. Illite increases the water stability of silt aggregates only up to 16% of the mineral content. The overall effect on water stability is the highest for kaolinite, then for montmorillonite and zeolite. The smallest effect is observed for goethite and illite, however, the initial impact of illite on aggregate stability is high, comparable to that of kaolinite. It appears that the dominant mechanism of the studied aggregates’ destruction is differential swelling, as far as breakdown by this mechanism increases with increasing clay content [13,39]. Aggregates containing larger doses of illite may break by slaking, as the slaking decreases with an increase in clay content [13].

It was assumed that the first candidate to be responsible for the time of aggregate destruction is the contact angle: a higher contact angle would retard the water penetration more, and therefore, the aggregate destruction. The dependence of the contact angle on the minerals’ content is presented in Figure 6.

It appears that among all studied minerals, kaolinite increases the water contact angle and water repellency to the greatest extent. Contrary to the above results, McKissock et al. [25,26] found that kaolinitic clays were the most successful in reducing water repellency in sandy soils. This discrepancy is not clear for the authors. By comparing Figure 5 and Figure 6, it becomes evident that the contact angle has little in common with the aggregate destruction time, which is particularly evident at low mineral rates: an increase in the destruction time in the range from 0 to 16% of the minerals is accompanied by a decrease in the contact angle. Most probably, the contact angle measured by the goniometric method is not a real one (despite attempts to level the material surface by applying very high pressure), and it is lowered by the pellet surface porosity, as was shown by Nosonovsky and Bhushan [40] for wetting liquids. Moreover, as was found by Hajnos et al. [41], the contact angle in mineral soils does not correlate with clay content (nor with organic matter content).

The dependence of the *WDPT* on the mineral content in the studied silt-mineral aggregates is shown in Figure 7.

The water drop penetration time increases with kaolinite and zeolite content in the whole experimental window. For the other minerals, it increases up to a 32% mineral percentage (64% for goethite) and then slightly drops further. Similarly, Dlapa et al. [42] found that *WDPT* was higher for higher clay contents in the majority of their measurements, however, they did not find a satisfactory explanation for this phenomenon. It can be caused, to a large extent, by a decrease in pore dimensions of the aggregate body due to large pores’ plugging by fine mineral particles. The measured infiltration times could also be affected by pressing. As reported by Bryant et al. [43], under compaction of water-repellent soil at various pressures (much lower than that used in this paper), *WDPT* markedly decreased with increasing pressure, however, wettable soil exhibited average *WDPT* values, similar to those of uncompacted soil. Apparently, the water drop penetration time is better related to aggregate destruction than the contact angle: *WDPT* correlates quite well with the aggregate destruction time, as shown in Figure 8.

For those who attribute *WDPT* to the wettability, the above correlation may indicate that more hydrophobic aggregates are more resistant to fast wetting. However, in the authors’ opinion, the *WDPT* has nothing to do with the wettability of hydrophilic beds and the above correlation indicates only that the porosity-dependent rate of water penetration into the material is a primary factor governing aggregate destruction.

As it is shown in Figure 4, for some aggregates, after initial rapid bubbling, a period of no bubbling occurs. This apparent hydrophobicity period (*AHP* (s)) depends on the type of mineral used and its concentration: except for illite aggregates, it occurs mainly at low mineral contents. The time-length of *AHP* is presented in Figure 9.

The phenomenon of apparent hydrophobicity may be connected to the wavy (sinusoidal-like) shape of pores. The dominance of wavy pores is very likely in the studied silt-mineral aggregates, particularly at low mineral concentrations. In this region, large skeletal pores between silt particles are only partially filled by fine particles of the minerals, and the wavy pore shape remains until the skeletal pores are filled completely or almost completely by the clay-size mineral phase. As it was theoretically explained by Czachor [44], for the wavy pore, the effective value of the wetting angle during liquid migration results from the real contact angle of the pore surface and the pore curvature at the contact point. At a certain point of the wavy pore wall (even if it is wettable), the capillary pressure may be negative, which means that external pressure is needed to push a liquid inward. The other possibility to allow the further penetration of water into the pore system is a destruction of the pore walls at the contact zone, which occurs in the aggregates studied.

The relation of the total volume of air released from an aggregate after immersion in water (volume of the bubbles) and the total volume of air present initially within the aggregate is shown in Figure 10. The latter value has been calculated either from bulk density (*BD*) or measured by mercury intrusion porosimetry (MIP).

The air (pore) volume measured by MIP correlates better with the volume of the bubbles, most probably because the aggregate surface is rough and contains large cavities, which are included in the pores calculated from *BD* and not measured by MIP. These surface cavities can be flooded by water without bubbling. At the first glance, the total volume of the bubbles should be equal to the total volume of air present within the aggregate, which means that the regression lines (or at least those for MIP) in Figure 10 should have the slope equal to 1 and 0 intercept. In both cases, the slopes are close to 1, but the intercepts are significantly smaller than 0. The latter means that statistically, around 0.6 cm^3^ of the air present within each aggregate is not measured by the air-bubbling method. Several reasons may be responsible for this effect:(1)Some water remains adsorbed on aggregate components’ surfaces at 60% humidity. As it can be read from adsorption isotherms, the thickness of the adsorbed layer at this humidity is around 2–3 molecules of water [45], so an amount of water present on one square meter of the adsorbent is around 3 × 10^−4^ cm^3^g^−1^ per gram. This is a negligible amount even for the studied pure montmorillonite, with a surface area of 200 square meters per gram.(2)Some bubbles released before the first registration point are not registered. This effect may be important, but only for aggregates exhibiting rapid initial flooding of funnel pores (i.e., for aggregates of low mineral percentages). Since the amount of air registered in the first second of the bubbling may be very high (even up to 0.6 cm^3^ for 2% kaolinite aggregate, usually less), and assuming the uncertainty of time reading of 0.5 s (one second was a period of time registration during the first few seconds), the nonregistered bubbling may account for up to 0.3 cm^3^.(3)Some air remains adhered to the destruction products.(4)Some air is released directly to the atmosphere due to water replacement (mainly from large pores) during aggregate immersion before its complete flooding.

In the authors’ opinion, the detailed balance of bubbles may serve for the estimation of a number of different pores, as well as of the rate of water penetration into fast wetted aggregates. It is worth examining in future experiments.

Bartoli et al. [46] described the water stability of the aggregate as a function of its tensile strength, which measures cohesion for air-dried aggregates. They confirmed it for Oxisol aggregates rich in kaolinite and goethite, but not Oxisol aggregates rich in gibbsite. For the aggregates studied in this paper, a similar dependence is presented in Figure 11. The values of the tensile strength of cylindrical aggregates made from the same materials were taken from [36].

The tensile strength correlated with the water stability (time necessary for complete aggregate destruction) for kaolinite-, montmorillonite-, zeolite- and goethite-containing aggregates, however no correlation was found for illite aggregates. A satisfactory explanation for this phenomenon was not found. It is possible that the correlation between mechanical and water stability occurs for these aggregates, which are destroyed in water by differential swelling (most of the studied minerals) and not by slaking (illite).

## 4. General Remarks

As compared to experiments performed on natural soils, studies of artificial aggregates of defined composition minimise the material heterogeneity that assures more correct interpretation and comparison of the results. The value of such results lies not in their direct transferability to soil systems but rather in obtaining more information on the mechanisms of aggregates’ stability.

The present study revealed that the effect of clay-size minerals on the water stability of silt aggregates decreases in the order: kaolinite, montmorillonite, zeolite, illite and goethite. However, the opposite sequences have been frequently reported in the literature. Such discrepancies are most probably caused by the fact that the reported data considered natural soil systems, where different synergic or antagonistic effects of various soil components play a significant role. Moreover, it is possible that the same minerals coming from different sources have quite different properties. It could be interesting to study this problem.

It seems interesting why the water stability of pure mineral aggregates is lower than that of the aggregates containing smaller (16–64%) proportions of the minerals. The authors suspect that it may be due to either the pore shapes of the aggregates or to the higher electrostatic repulsion between the same (minerals) than between different (silt and minerals) particles.

A question remains as to why the contact angle does not correlate with the aggregate water stability. Is it only a matter of the uncertainties of the contact angle measurements or are the other factors much more important?

## Data Availability

All data are available from the authors upon reasonable request.

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
