# Peer review of "Effect of Different Minerals on Water Stability and Wettability of Soil Silt Aggregates"

_materials, 2022, doi:10.3390/ma15165569_

Round 1

Reviewer 1 Report

 You can find my suggestions and corrections are below:

1.      Please add the information about some devices in the text (model, brand, country etc.) In line 94, add the information for pressed pellet, and in line 106, add the information about mercury intrusion porosimetry (MP).

2.      Interpretation of the graphics for each mineral in Figure 4 can be added in the text briefly.

3.      Why the reference in line 223 have different results from your study? Can you explain the difference of studies and results.

4.      There is no conclusion for general results. Please add it after results.

Author Response

Dear Reviewer 1,

please find below our answers for your comments. The added text referring to them (and to the other Referees suggestions, as well) is marked red in the corrected manuscript.

  1. Please add the information about some devices in the text (model, brand, country etc.) In line 94, add the information for pressed pellet, and in line 106, add the information about mercury intrusion porosimetry (MP).

All information was added in the text.

  1. Interpretation of the graphics for each mineral in Figure 4 can be added in the text briefly.

We added a small interpretation of the data below the Figure. Since the detailed interpretation of all graphics has been provided in the text below, we did not like to enlarge this fragment not to repeat the text.

  1. Why the reference in line 223 have different results from your study? Can you explain the difference of studies and results.

It was difficult for us to explain this discrepancy. We mentioned it in the text. We did not like to speculate on the possible reasons (McKissock used sand and we used silt, different additional components could be present in both systems, the kaolin of different origin and properties might have been used in both papers). By the way it could be interesting to study the action of the same mineral coming from different sources. We commented it in General remarks section.

  1. There is no conclusion for general results. Please add it after results.

Since the Editorial policy is not to write Conclusions because they usually repeat the Abstract and the text, we added “General Remarks” section containing some more general view on the results.

Yours sincerely, the authors.

Reviewer 2 Report

This paper conducted water stability and wettability of artificial soil to find out some interesting phenomena. Overall, it is a meaningful paper, but some major issues remain as follows.

1 Why the title of this article has “.”?

2 Where is the conclusion part?

3 There is too much content in Section 2.2 of “Studies of the aggregates”, some of which should remove to Section 3.

4 Soil composition varies from place to place. How to ensure the representativeness of this research?

5 Except for the line charts, the readers also want to see some images directly related to the research results.

Author Response

Dear Reviewer 2,

please find below our answers for your comments. The added text referring to them (and to the other Referees suggestions, as well) is marked red in the corrected manuscript.

  1. Why the title of this article has “.”?

The title was simplified to “Effect of different minerals on water stability and wettability of soil silt aggregates”. Previously we wanted to express that we observed a period of apparent hydrophobicity, that, according to our knowledge, has not been explicitly reported up to date.

  1. Where is the conclusion part?

Since the Editorial policy is not to write Conclusions because they usually repeat the Abstract and the text, we added “General Remarks” section containing some more general view on the results.

  1. There is too much content in Section 2.2 of “Studies of the aggregates”, some of which remove to Section 3.

Really the content of this section seems to be too big. At first we tried to move the whole calculations (Figs 1-3 + the respective comments) below Fig 4, then we moved only Figs 2 and 3, however in both cases the whole text and the explanations were not smooth. Here we have in mind that to describe the curves in Fig 4 some terms (i.e. funnel pores flooding, apparent hydrophobicity and “pure destruction) are needed which in the case of the text rearrangement would be explained later than needed. Of course if you insist to rearrange the text, we will do it.

  1. Soil composition varies from place to place. How to ensure the representativeness of this research?

You are right, soil heterogeneity is usually the main factor which affect incompatibility of the results obtained by different Authors on even the same soil types. Compared to field experiments, our approach minimizes soil heterogeneity that assures a correct interpretation and comparison of the results. The validity of such results lies not in their direct transferability to soil systems but rather in getting information on mechanisms of aggregates stability. We used this answer in General remarks section.

  1. Except for the line charts, the readers also want to see some images directly related to the research results.

The images presented the example of water drop behavior during contact angles and WDPT measurement was added to the manuscript. Unfortunately we did not film the processes of the aggregates destruction. To do this we have to produce new aggregates and dry them. This is time-consuming and it is not possible to be done before the Editor’s deadline to complete the review.

Yours sincerely, the authors.

Reviewer 3 Report

In this paper, the effect of different minerals on water stability of soil aggregates were investigated by observation of apparent hydrophobicity period during fast wetting. In general, this paper is interesting and is in the journal scope. However, the paper was not well written and the methodology should be clearly presented and explained. Therefore, this article should be majorly revised before publication. Some comments can be found as follows:

(1) The first person voice, such as we, I, our, us, is often not used in scientific papers. Please check the full text to remove first person voice.

(2) In the section of abstract, the important results and conclusions obtained from this study should be pointed out besides the sentific work conducted by the authors.

(3) In the section of introduction, at the end of literature review, the main problems and shortcomes in the current study should be summarized based on the literature review. And then the main Innovative work can be proposed based on this.

(4) The properties of the main materials used in this study should be presented.

(5) The experiment method should be illustrated in details.

(6) After the section of Results and Discussion, there should be added a section of Conclusions, in which the main conclusions should be summarized.

(7) The references cited in this paper are too old. Some updated literatures published in the past three years should be added.

Author Response

Dear Reviewer 3,

please find below our answers for your comments. The added text referring to them (and to the other Referees suggestions, as well) is marked red in the corrected manuscript.

  1. The first person voice, such as we, I, our, us, is often not used in scientific papers. Please check the full text to remove first person voice.

The first person voice was removed from full text of manuscript.

  1. In the section of abstract, the important results and conclusions obtained from this study should be pointed out besides the scientific work conducted by the authors.

We enlarged the Abstract according to your request.

  1. In the section of introduction, at the end of literature review, the main problems and shortcomes in the current study should be summarized based on the literature review. And then the main innovative work can be proposed based on this.

Now we explicitly wrote that the main innovative problems we wanted to solve in the current study were to observe the “clear” effects of pure minerals on water stability of aggregates to eliminate the influence of various factors occurring in natural soils and to apply a method which allows for such studies, i.e. is enough precise to study rapidly destroyed aggregates.

  1. The properties of the main materials used in this study should be presented.

There is an information (line 95) that the detailed description of the materials is presented in the reference 29. These basic properties are specific surface area (from nitrogen adsorption), average particle diameter, zeta potential, volume of 10–30 nm micropores, average pore diameter, solid phase density, surface fractal dimension, volumetric porosity (fraction of pores), average pore diameter to average particle diameter ratio). It is mentioned in the text.

  1. The experiment method should be illustrated in details.

As it was mentioned in line 113 water stability of the aggregates was estimated from air bubbling after immersion using a method described in the paper by Jozefaciuk et al. [27], where all detailed information about the experimental method, illustrations etc. are given.

  1. After the section of Results and Discussion, there should be added a section of Conclusions, in which the main conclusions should be summarized.

Since the Editorial policy is not to write Conclusions because they usually repeat the Abstract and the text, we added “General Remarks” section containing some more general view on the results.

  1. The references cited in this paper are too old. Some updated literatures published in the past three years should be added.

The literature was updated – the papers from last three years were added to the text and reference list. In the view of above the numbers of references changed.

Yours sincerely, the authors.

Reviewer 4 Report

The authors studied the water stability of artificially prepared soil slit aggregates containing 5 different minerals. They applied a new recently published method for this aim. In addition, they measured the initial contact angle and the time for penetration of droplet. The rate of aggregate destruction in water increased with increasing mineral content. The easy wetted aggregates are destroyed faster. The work is technically well conducted and well written. I recommend its publication in the present form.

Author Response

Dear Reviewer,
Thank you very much for your time and attention to our manuscript

Yours sincerely, the authors.

Round 2

Reviewer 1 Report

The required changes have been made by authors. The manuscript is suitable for printing.

Reviewer 2 Report

The authors have revised the manuscript to meet the journal requirement .

Reviewer 3 Report

Thanks for your revision. The manuscript can be accepted for publication in its current version.